# Attenuation of HIV severity by slightly deleterious mutations can explain the long-term trajectory of virulence evolution

Harriet Longley[1,2], Christophe Fraser[1,2], Chris Wymant[1,2], Katrina Lythgoe[1,2,3]*

**1** Big Data Institute, Li Ka Shing Centre for Health Information and Discovery, Nuffield Department of Medicine, University of Oxford, Oxford, United Kingdom, **2** Pandemic Sciences Institute, Nuffield Department of Medicine, University of Oxford, Oxford, United Kingdom, **3** Department of Biology, University of Oxford, Oxford, United Kingdom

* katrina.lythgoe@biology.ox.ac.uk

## Abstract

HIV-1 is a well-studied example of a pathogen that has evolved an intermediate level of virulence that maximises transmission. For a trait to evolve it must be heritable, and although viral load—a proxy for disease severity—has been shown to be a heritable trait, it is surprising that specific heritable viral factors remain mostly elusive. Rapid within-host evolution is also expected to diminish heritability. We hypothesised that rather than a small number of mutations of large effect determining viral load, the number of slightly deleterious mutations could be key. As a proof of principle, we explored how viral load is expected to evolve within and between hosts using a nested modelling approach that links within-host evolution with epidemiological outcomes. For mutations of sufficiently small effect, a mutation-selection balance is gradually reached during infection, resulting in slow changes in viral load despite rapid rates of genomic evolution. In simulated epidemics, we generated realistic population distributions of viral loads and estimates of heritability. The existence of many slightly deleterious mutations provides a mechanism that can help to explain why viral loads change slowly during infection, broad distributions of viral loads among individuals, and why searches for viral factors that determine viral load have had limited success.

## Author summary

HIV-1 evolves rapidly, yet tends to maintain an intermediate level of virulence—measured by viral load—that optimizes transmission. While viral load is heritable, the specific viral genetic factors involved have remained elusive. We hypothesized that this is due not to a few major mutations, but to the gradual accumulation of many slightly harmful (deleterious) mutations. To test this, we

**Data availability statement:** Since this study is purely theoretical, no empirical datasets were analyzed. However, scripts used to generate synthetic data from the models are available in the GitHub repository: https://github.com/HLongleyOx/EvolutionOfVirulence2025.

**Funding:** H. Longley is supported by the Engineering and Physical Sciences Research Council Centre for Doctoral Training in Health Data Science (EP/S02428X/1). K. Lythgoe is supported by the Royal Society and the Wellcome Trust (107652/Z/15/Z) and the Li Ka Shing Foundation. C. Fraser is supported by the Li Ka Shing Foundation. The funders had no role in study design, data collection and analysis, decision to publish, or preparation of the manuscript.

**Competing interests:** The authors have declared that no competing interests exist.

developed a nested model linking mutation dynamics within individuals to viral load patterns across populations. We systematically varied the number and strength of mutations to explore how these factors influence both within-host evolution and between-host transmission. Our findings reveal that when many mutations each have a small negative effect, they reach a mutation-selection balance. This balance slows changes in viral load over time, even though the virus continues to evolve genetically. This mechanism helps explain why viral loads remain relatively stable within individuals, vary widely across the population, and why viral genetic associations are hard to detect. We also show that host genetic diversity increases this variation. Together, our results suggest that weakly deleterious mutations play a key role in HIV virulence evolution, offering new insight into how HIV adapts—and how it might be better controlled.

## Introduction

HIV-1 is an example of a pathogen that has evolved towards intermediate virulence as a result of the trade-off between virulence—defined here as disease severity—and transmission [1]. The trade-off hypothesis [2,3] proposes that while higher virulence can increase transmission rates, it can also reduce the duration of infection by compromising host survival, thereby limiting the overall transmission potential. If the relationship between virulence and transmission rate saturates, the transmission fitness of the pathogen will be maximised at a finite level of virulence [4]. If in addition virulence is heritable, meaning that differences in virulence between individuals are partly determined by genetic differences in the infecting pathogen, the pathogen is expected to evolve towards the level of virulence that maximises the number of onward transmissions.

Demonstrating the existence of evolutionary trade-offs of pathogens in real-world systems is notoriously difficult [4–6], and HIV-1 is one of the few systems where a transmission-virulence trade-off has been shown [1]. During untreated chronic infection, HIV-1 viral loads typically remain at a relatively steady level, known as the set-point viral load (spVL). This is a striking observation given that spVLs are extremely heterogeneous when measured among individuals, varying by over four orders of magnitude [7]. Because untreated individuals with higher viral loads tend to progress to AIDS and death sooner than those with lower viral loads [7,8], spVL is the most commonly used proxy for HIV-1 virulence. In addition, viral load has been shown to be correlated with infectiousness, with high viral load individuals much more likely to transmit the virus per contact [9,10]. Fraser et al. [1] showed that the relationship between the expected number of onward transmissions and the duration of infection is saturating, and moreover that observed viral loads cluster around the values that are expected to maximise transmission, thus supporting the trade-off hypothesis. In a follow-up study, Blanquart et al. [11] found further support for the trade-off hypothesis from a large HIV-1 prospective cohort and argued that the attenuation of viral loads in the cohort over two decades was the result of between-host adaptation towards maximising transmission potential.

For a trait to evolve under natural selection it must be heritable. Although the concept of heritability has its roots in animal breeding, the same concept can also be applied to pathogen virulence, i.e., for measuring the extent to which severity of infection is determined by the genotype of the infecting pathogen. Once accounting for differences in datasets and methodology, broad-sense heritability of spVL is estimated to be between 20–40% across studies [12–18]. Reported broad-sense heritability estimates can differ depending on the evolutionary model assumed: neutral (Brownian motion) models, which treat variation in spVL as drifting randomly across the phylogeny, yield lower values (e.g., 5.7% in Hodcroft et al. [19] 8% for Brownian motion in Bertels et al. [15]), whereas stabilising (Ornstein–Uhlenbeck) models, which assume selection toward an intermediate optimum consistent with the transmission–virulence trade-off and are more biologically realistic, have produced higher estimates (e.g., 29% for the OU model in Bertels et al. [15]). These differences reflect both model choice and whether estimates are adjusted for covariates such as age and sex. In contrast, genome-wide association studies (GWAS) have provided narrow-sense heritability estimates, which quantify only the additive effects of identified variants and are typically adjusted for demographic covariates. As such, GWAS estimates are not directly comparable to broad-sense phylogenetic estimates but are most reasonably considered alongside covariate-adjusted broad-sense estimates. Studies seeking to identify individual viral mutations associated with differences in viral load have rarely identified single variants [17,20]. Instead, associations are highly polygenic and dispersed across the genome, and even significant sites explain only a small proportion of the observed variance.

Host-factors have also been connected to variation in viral loads [21]; however, a 2017 study found the fraction of variation explained by human genetic factors to be relatively low at 8.4% once viral genetic diversity is accounted for [20]. Some human polymorphisms, such as the 32 base pair deletion in the CCR5 gene, have been found to significantly reduce HIV virulence [22,23]. Virus-host interactions also likely affect virulence, for example the changes in immune escape pressure following transmission to a new host environment [24].

Another difficulty in understanding the heritability of HIV-1 spVL is rapid within-host evolution and often long delays between infection and onward transmission. HIV-1 has a high mutation rate per replication which, coupled with a short viral generation time, generates significant genetic diversity within individuals and the potential for rapid adaptive evolution [25]. This process is likely to reduce heritability as the viral genotype an individual is infected with is expected to differ from the genotype that they go on to transmit [18,26]. For example, we may expect viral variants with high replicative capacity to emerge and quickly sweep through the virus population, and if replicative capacity affects disease factors such as virulence, then the heritability of these factors will be reduced.

Viral replicative capacity has been shown to be associated with viral load [27–29]. Even fairly small differences in the fitness of virus variants are expected to result in rapid within-host evolution, notwithstanding complications of whether the virus is pre-adapted to a recipient's HLA allele type [30,31]. Hence if there were a strong link between replicative capacity and viral load, we would expect viral loads to increase greatly during infection, low estimates of heritability, and the evolution of high virulence even at a cost to overall transmission ("short-sighted evolution") [24,32]. Yet viral loads only change modestly during chronic infection [33,34], heritability of viral load is remarkably high, and ever-increasing viral loads has not been a feature of the HIV-1 epidemic. Possible mechanisms for how virulence has evolved under the constraints of both within and between host selection include a complex fitness landscape, and the preferential transmission of ancestral strains [18]. Underpinning any mechanism are the elusive heritable viral factors.

We propose that rather than being controlled by few high-impact mutations (as usually targeted by whole-genome association studies), viral load is largely determined by the number of slightly deleterious mutations a genome has. In such a scenario a mutation-selection balance, in which mutations continually arise through mutation and are slowly lost through purifying selection, is gradually approached during the course of infection. Because this process is inherently slow, it can reconcile seemingly incompatible aspects of the within- and between-host processes, namely the stability of viral load during chronic infection, the high heritability of viral load, and the selection of intermediate viral loads at the population level. Moreover, it may explain why HIV-1 virulence factors have been so hard to identify. As a proof of concept, we

considered the impact of many slightly deleterious mutations on the within and between-host evolutionary dynamics of the virus within treatment-naïve individuals using a nested modelling approach.

## Methods

We modelled the evolving population of viral genotypes during infection as a system reaching mutation–selection balance by utilising the quasispecies framework. Mutations occur at a rate of $\mu = 3 \times 10^{-5}$ per site per generation [35] at $m$ sites only, with the remainder of the genome ignored. We assume each of the $m$ sites in the viral genome has two potential alleles, wild-type or slightly deleterious, with 'virus types' defined by the number of deleterious mutations in the viral genome, regardless of the position of those mutations, and hence viruses with different genotypes can have the same virus type. Individuals are assumed to be initially infected by a single virus type, which defines the infection type: an individual who was infected with virus type $j$ will be of infection type $j$. After infection, the dual forces of a high mutation rate and weak selection result in the emergence of a diverse within-host viral population and the maintenance of deleterious mutations, with the population reaching mutation-selection balance. At time $t$ during a type $j$ infection, a virus type $i$ has a frequency $x_{ij}(t)$ that is determined by the quasispecies equation (see next sub-section). We assumed that the viral load of an infection at time $t$ is determined by the number of deleterious mutations in the within-host viral population at time $t$. For the epidemiological (between-host) modelling, we use a nested modelling framework from Lythgoe *et al*. [32] in which the within-host dynamics are 'nested' within a between-host epidemiological model, which in turn is based upon the theory of multi-type epidemic models [36], and the course of an individual's infection is entirely determined by the within-host model.

### Within-host dynamics

We modelled within-host virus evolution by considering a simplified genome of $m$ segregating sites, representing positions of mutations with a deleterious effect of viral fitness, which in turn lowers the associated viral load. This abstraction allows us to focus on the evolutionary dynamics of viral load with a quasispecies framework without modelling the full HIV genome. Each virus type is therefore defined by the number of deleterious mutations $j \in \{0, 1, \ldots, m\}$, without regard to their specific positions.

We assumed that each newly infected cell inherits a virus type from the infecting virion and may acquire or lose at most one mutation during replication. The mutation process is described by an $(m + 1) \times (m + 1)$ matrix $Q = [q_{ij}]$, where $q_{ij}$ is the probability that a progeny virion is of type $i$, given a parental type $j$. Specifically, the entries of $Q$ are defined as:

1. Forward mutation (type $j \to j + 1$) : $q_{j+1,j} = \mu \cdot (m - j)$

2. Backward mutation (type $j \to j - 1$) : $q_{j+1,j} = \mu \cdot j$

3. No mutation (type $j \to j$) : $q_{j+1,j} = 1 - \mu \cdot m$

We considered the competing virus types within a host as a quasispecies for which the dynamics are governed by the quasispecies Equation (1), and the relative fitness of virus type $j$ is a multiplicative function of the number of deleterious mutations: $A_j = (1 - s)^j$ where $s$ is the selection coefficient and $A_0 = 1$. Here, viral fitness is synonymous with the replicative capacity of the virus type. The change over time in the relative frequencies of the virus types can be described by the quasispecies equation [38]:

$$\frac{dx}{dt} = Wx - \overline{w}x$$

where $W = w_{ij} = q_{ij}A_j$ and the term $\overline{w} = \Sigma_i^m \Sigma_j^m w_{ij}x_{ij}(t)$ bounds the sum of the frequencies to one. The solution at equilibrium, $\widetilde{x}$, can be found analytically as the dominant eigenvector of $W$ [37]. The pre-equilibrium analytical solution of the system was solved numerically with the *deSolve* package in R v. 4.0.2.

The higher the fixed number of segregating sites in the model, the smaller the relative fitness cost of one additional mutation. We explored a range of fitness costs, from $10^{-2}$ per mutation when 10 sites are segregating to $5 \times 10^{-5}$ per mutation when 250 sites are segregating, spanning strong selection to close to neutral effects approaching the mutation rate [38]. We capped the maximum number of segregating sites, $m$, at 250 because the matrices needed to track all possible combinations of viral variants become too large to handle - they grow exponentially with each additional site we add to the model - however the range of values we considered effectively capture the key dynamics of the model.

## Viral load

There are three distinct stages of an HIV infection. The first and third stages of infection, termed acute and late infection respectively, are characterised by high viral loads. The second stage is chronic infection, when viral loads are relatively stable yet vary significantly between individuals. We assumed that during chronic infection the viral load at a given time is determined by the current composition of the viral population. As a consequence, the viral load can change as the viral population evolves: the fewer the number of deleterious mutations the higher the viral load. At any moment in time the viral population is likely to be comprised of multiple types, so we first define the contribution of each viral type, $V_i$, to the viral load during chronic infection:

$$V_i = V_0 - \lambda_m i$$

Where $V_0 = 7 \log_{10}$ (viral copies per ml), $i$ is the number of mutations, and $\lambda_m$ is the reduction in viral load due to an additional mutation, which is in turn determined by the maximum number of mutations, $m$. The maximum possible viral load, $7 \log_{10}$(viral copies per ml), was chosen to match the highest viral loads typically observed [18], and similarly the lowest viral load is fixed at $2 \log_{10}$(viral copies per ml).

When we assumed a large number of mutations, we assumed that the fitness cost of a single mutation is small, and therefore the reduction in viral load given one additional mutation is also small. Conversely, if we have few mutations of large effect, we assumed viral load will reduce significantly when a deleterious mutation appears. The value of $\lambda_m$ is chosen such that the difference in the viral load between the fittest and weakest virus is $5 \log_{10}$(viral copies per ml) to correspond with a realistic range of viral loads, and so $\lambda_m = \frac{5}{m}$, meaning that a virus type with the maximum number of mutations had an associated viral load of $2 \log_{10}$(viral copies per ml) in agreement with the observed ranges of spVLs in chronic infection. In determining viral load in this way, there is a linear relationship between the viral load - $\log_{10}$(viral copies per ml) - of a variant and the relative fitness of the variant. We then determined the realised viral load to be the mean of these contributions, so that viral load of a type $j$ infection at time $t$ into infection is given by:

$$V_j(t) = \Sigma_{i=0}^{m} x_{ij}(t) V_i$$

## Duration of chronic infection

We used the previously parameterised decreasing Hill function for the duration of chronic infection as a function of spVL [1]. To account for the change in viral load over time in the calculation of the duration of infection, we took a weighted average of the viral load over the maximum possible duration of chronic infection, determined to be 20.4 years by the Hill function. The weight of $V_j(t)$ in the calculation of $\overline{V}_j$ is inversely proportion to $t$, such that the viral load close to the start of chronic infection contributes more to the calculation than the viral load several years into infection. The weighted average viral load is then used as an input to the decreasing Hill function to provide a chronic infection duration, termed $T_j$. To attribute a spVL, $V_j$, to an infection we take the arithmetic mean viral load over $T_j$.

## Infectivity profile

In order to determine how evolution proceeds at the between-host level, we need to know not only the frequency of different virus types during infection, but also their probability of transmission. We assumed a single virus type is transmitted during a transmission event, and the hazard for transmitting virus type $i$ in a type $j$ infection at time $t$ is:

$$\beta_{ij}(t) = x_{ij}(t)\gamma_j(t) , \qquad t < T_j$$

$$\beta_{ij}(t) = 0 \qquad \text{Otherwise}$$

Where $\gamma_j(t)$ is the infectiousness of a virus type $j$ infection at time t and $T_j$ is the duration of a virus type $j$ infection. For all infections we assume the acute phase has a duration of 0.25 years and an infectiousness of 2.76 onward infections/year, and the late phase has a duration of 0.75 years and an infectiousness of 0.76 onward infections/year [1]. During chronic infection, the infectiousness $\gamma_j(t)$ is described by an increasing Hill function of viral load, $V(t)$, at time $t$ with parameter values based upon the optimal values detailed in [1] (Table 1).

## Between-host model

We used an SI model with demography to model between-host transmission, where we assumed a natural death rate $D$ and that individuals enter the susceptible pool of individuals at a rate of $B$. The force of infection of virus type $i$ at time $\tau$ of an infection founded by virus type $j$ is defined as $\beta_{ij}(\tau)e^{-D\tau}$ when $\tau \le T_j$ and 0 otherwise. The between-host dynamics are modelled by the renewal equation, as described in Lythgoe *et al.* [32]. Specifically, incidence is calculated by the renewal equation which follows the logic that incidence at time $t$ is the integral of the past incidences weighted by how much the individuals previously infected would still be transmitting. The between-host dynamics at time $t$ since the beginning of the epidemic are therefore described as follows:

$$I_i(t) = \frac{S(t)}{N(t)} \sum_{j=1}^{m} \int_0^{T_j} \beta_{ij}\, I_j(t-\tau)e^{-D\tau}d\tau$$

$$S(t) = N(t) - \sum_{i=1}^{m} \int_0^{T_j} I_j(t-\tau)e^{-D\tau}d\tau$$

$$\frac{dN(t)}{dt} = B - DN(t) - \sum_{i=1}^{m} I_i(t-T_i)e^{-DT_i}$$

Where $N$ is the total population size, $S$ is the number of susceptible individuals and $I_i$ is the incidence of infections founded by a type $i$ virus. All individuals have the same natural death hazard, $D$, and infected individuals also have an infinite death hazard at the moment their infection ends, which is pre-determined by their infection type.

We began the epidemic with an incidence of 1. The solutions were determined numerically using the basic forward Euler method. We ran 3 simulations of the between-host dynamics, where the initial circulating virus type and therefore average viral load differed for each simulation. Epidemiological theory shows that the next generation matrix $K$, with $ij$th element $k_{ij} = \int_0^{T_j} \beta_{ij}(\tau)e^{-\mu\tau}d\tau$, has a unique and real dominant eigenvalue that gives the value of the basic reproduction number $R_0$, and the associated eigenvector describes the population structure of the virus types at equilibrium [32], which

**Table 1. Variable and parameters descriptions for the within-host and between host models.**

| Parameters | Definition | Value |
|---|---|---|
| $\mu$ | Mutation rate | $3 \times 10^{-5}$ (35) |
| $m$ | The number of segregating sites, and therefore the maximum number of deleterious mutations. Six values were explored in-depth. For the within-host equilibrium solution, the solution was calculated for a large number of values, ranging 10 to 400 in increments of 10. | 10, 50, 100, 150, 200, 250 |
| $s_m$ | The relative fitness cost of an additional mutation ranging from $s_{10} = 10^{-2}$ to $s_{250} = 5 \times 10^{-5}$ to capture a realistic distribution of selection coefficients. The values between m=10 and m=250 were determined by fitting a straight line between $s_{10}$ and $s_{250}$ on a log-linear scale to scale selection strength with the number of segregating sites. | $s_{10} = 10^{-2}$ <br> $s_{50} = 10^{-2.4}$ <br> $s_{100} = 10^{-2.9}$ <br> $s_{150} = 10^{-3.4}$ <br> $s_{200} = 10^{-3.8}$ <br> $s_{250} = 10^{-4.3}$ |
| $A_j$ | Replication rate of a type $j$ virus variant. | $A_j = (1-s)^j$ |
| $\lambda_m$ | The cost of an additional mutation on the associated viral load. It is given by $\lambda_m = \frac{5}{m}$. The value is determined such that the minimum viral load is $2\log_{10}$(viral copies per ml). | $\lambda_{10} = 0.5$ <br> $\lambda_{50} = 0.1$ <br> $\lambda_{100} = 0.05$ <br> $\lambda_{150} = 0.033$ <br> $\lambda_{200} = 0.025$ <br> $\lambda_{250} = 0.02$ |
| $T_j$ | The duration of a type-$j$ infection. | Variable |
| $\gamma_j(t)$ | Infectivity of a type-$j$ infection at time $t$ into infection. | Variable and a function of $V_j(t)$. Parameterised in [1]. |
| $V_i^s$ | The viral load associated with a type-$i$ virus for a given fitness cost. | $7 \log_{10}$(copies per ml) $- \lambda_s i$ |
| $B$ | Rate at which individuals enter the susceptible population. | 200 per year [32] |
| $D$ | Natural mortality rate. | 0.02 per year [32] |
| $h$ | Host type. | 1-50 |
| $e$ | Maximum absolute host effect on viral load on $\log_{10}$ scale | 0.1, 0.25, 0.5, 0.75, 1 |
| $E_h$ | Additive host effect size on $\log_{10}$ viral load. | Discretely uniformly distributed over the range [-e, e] in the host population. |
| Variables | | |
| $x_{ij}(t)$ | The frequency of a type-$i$ virus at time $t$ of a type-$j$ infection. | |
| $V_j(t)$ | Viral load at time t of a type-$j$ infection | |
| $\beta_{ij}(t)$ | The infectivity profile, i.e., the rate at which a type-$i$ virus is transmitted at time t of a type-$j$ infection. | |
| $K$ | The next generation matrix. | |
| $I_i(t)$, $I^*$ | The number of infected individuals infected with a type-$i$ virus at time $t$ into the epidemic ($I_i(t)$) and at the endemic equilibrium ($I^*$). | Initially 1 |
| $S(t)$, $S^*$ | The number of susceptible individuals at time $t$ into the epidemic ($S(t)$) and at the endemic equilibrium ($S^*$). | Initially 10,000 |
| $N(t)$, $N^*$ | The number of human hosts at time $t$ into the epidemic ($N(t)$) and at the endemic equilibrium ($N^*$). | Initially 10,000 |
| $h^2$ | Broad-sense heritability estimate. | |

can be normalised to give the proportions of each virus type at their equilibrium state, $I_j^*$. The transmission potential of an infection by a type $j$ virus is defined as the number of expected onward transmissions during infection and is determined from the next generation matrix as $p = \Sigma_i k_{ij}$, where the term $k_{ij}$ is the expected number of transmissions of a type $i$ virus during an infection of a type $j$ virus.

## Host heterogeneity

As well as considering a population of genetically identical host individuals, we also examined the effect of host heterogeneity. The host effect is quantified by an additive effect to the viral load, such that for a type $j$ infection in a host type $h$, the viral load at time $t$ is given by $V_j^h(t) = V_j(t) + E_h$. The parameter $E$ is discretely uniformly distributed, $E \sim \mathcal{U}(-e, \ e)$ and there are 50 possible host types. As a result, the probability of transmitting virus type $j$ at time $t$ differs by host type, however the within-host dynamics are identical for all infections. We considered a range of values of $e$ from 0.1 to 1 in order to quantify the impact of host specific viral load effects on model outcomes. We present the analytical equilibrium solution for populations with host-heterogeneity as it was not feasible to derive pre-equilibrium solutions numerically due to computational constraints.

## Heritability

To estimate broad-sense heritability, $h^2$ (heritability hereafter for brevity), we use the classic parent-offspring regression method [39]. To apply the method to a simulated infection population, we generated 1000 samples of 500 transmission pairs, where the source infection type was sampled based upon the population distribution of infection types at the endemic steady state. The probability of a recipient having an infection of a type $i$ virus from a source infected with type $j$ virus was described by the vector of transmission potentials, $k_{ij}$, normalised to sum to 1. For each set of transmission pairs, an estimate of $h^2$ was given by the regression coefficient of a simple linear regression with recipient spVL as the outcome. The overall estimate of heritability is calculated by the average $h^2$ across the sets of transmission pairs. For a non-homogeneous host population, the host type of the source and recipient was also sampled from a uniform distribution of host types.

# Results

To summarise our approach, we modelled the within-host dynamics with a quasispecies model that describes the changing frequencies of competing virus types within a host virus population, where each virus type is defined by its number of deleterious mutations. The relative fitness of a virus type is a function of the number of deleterious mutations, and the conflicting forces of a high mutation rate and purifying selection results in a mutation-selection balance of deleterious mutations if the fitness cost is sufficiently low. The viral load at time $t$ is determined by the average fitness of the virus population at time $t$, where high fitness induces a high viral load. The within-host model is nested in a between-host model that can describe the population-level distributions of viral loads and the amount of heritability for different parameter choices.

## Many low-cost mutations slow within-host evolution and lead to intermediate virulence

We determined the within-host equilibrium solution for a range of values for the number of segregating sites, $m$, and calculated the corresponding viral load. Given our choice of model parameters, we find that for a relatively low number of sites (fewer than approximately 100), selection of the fittest virus types dominates, and viral load is at the maximum possible value. In our model, as the number of sites increases, the selection cost decreases, and the system approaches a mutation-selection balance. Consequently, the number of deleterious mutations in the population is maintained and the viral load is reduced. Eventually, the cost of an additional mutation passes a threshold where the relative fitness cost is close to neutral and the equilibrium viral load approaches the midpoint of the viral load range of $4.5 \log_{10}$(viral copies per ml), and the genetic variation is maintained through mutation (Fig 1).

With more segregating sites, the mutation-selection balance maintains a more diverse population, where multiple viral types coexist at equilibrium rather than a single dominant type (S1 Fig). Higher diversity in the within-host population creates a broader pool of variants available for transmission, which influences the trajectory of between-host evolution.

Adjusting how selection strength scales with the number of segregating sites shifts model outcomes. When strong selection (high $s$) acts across many sites, deleterious mutations are efficiently purged, maintaining higher equilibrium viral loads.

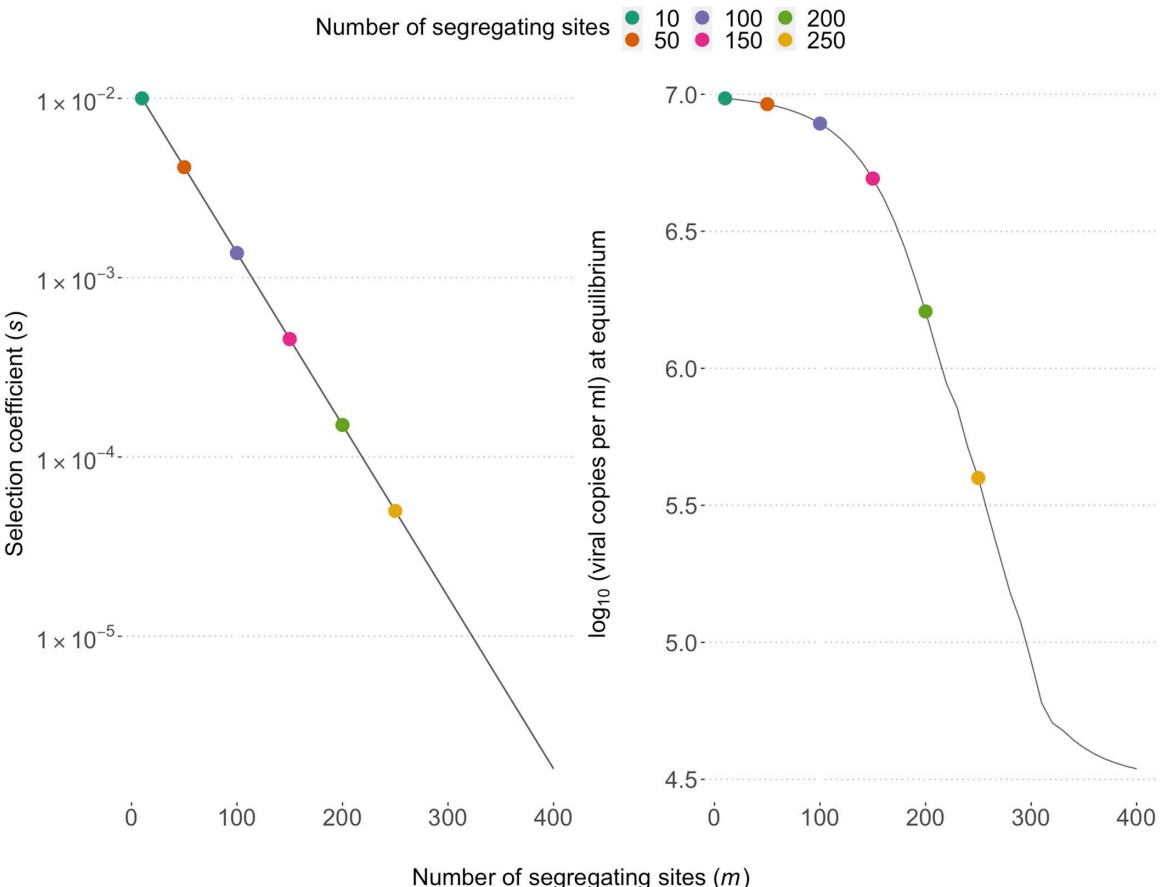

**Fig 1. Within-host model fitness and equilibrium viral load.** A) The relationship between the number of segregating sites and the fitness cost (synonymous with selection coefficient) associated with an additional mutation. We explore a range of values of $m$ (ranging 10 to 400 in increments of 10), with the corresponding fitness cost of a mutation reducing on a log-linear scale as $m$ increases. The six choices of m explored here are indicated by coloured points. B) The viral load at equilibrium of an infection with an increasing number of segregating sites. As the number of sites increase and the fitness costs decrease, the population at equilibrium is characterised by mutation-selection balance that maintains deleterious viral types in the population, thus lowering viral loads.

Conversely, if selection weakens as more sites accumulate, deleterious mutations build up, reducing equilibrium viral load. Varying $s$ at fixed $m$ ($m = 100$) demonstrates that dynamics slow as $s$ decreases (S2 Fig), and the average within-host viral load lowers at the equilibrium state. From a population genetics perspective, increasing $m$ raises the mutational load—the expected reduction in mean fitness from deleterious mutations—so even with constant $s$, average fitness declines as more sites become mutable. While specific equilibrium values depend on $m$ and $s$, the qualitative behaviour remains unchanged: altering $s$ at fixed $m$ affects the tempo of dynamics and final population-level fitness but not the overall pattern we report. Our focus is on the cumulative impact of many small-effect mutations, which can substantially reduce viral load, compared to fewer large-effect mutations; therefore, we emphasise scenarios where $s$ decreases as $m$ increases.

To explore the model and its dynamics, we considered six scenarios differing by the number of segregating sites (10, 50, 100, 150, 200, 250) and the fitness cost $s$ of each mutation. The viral load calculation ensures that the virus type with the maximum number of mutations $(m)$ has an associated viral load at the lower tail of reported spVLs, approximately $2 \log_{10}$(viral copies per ml), and the fittest virus has an spVL of $7 \log_{10}$(viral copies per ml) [1]. For each of the scenarios, we determined the evolutionary dynamics of the infection for all possible starting infection types. As expected, if there are

few segregating sites, each with mutations of large effect, a within-host equilibrium was rapidly reached within months and was dominated by the fittest variants harbouring no mutations ([Fig 2A]). Increasing the number of sites to 50 and 100 slows the within-host dynamics, however all infections have reached the maximum possible viral load within 5 years and 15 years respectively, with initially low viral loads steadily increasing throughout chronic infection.

When a large number (>100) of sites are segregating and fitness costs are lower we approach a mutation-selection balance, lowering the viral load. Moreover, it took significantly longer than the assumed maximum infection duration to approach this equilibrium, depending on the number of mutations of the infecting strain, and therefore at the end of the maximum infection duration (~20 years) there remains variation across infection types. As we continue to increase the number of segregating sites from 200 to 250, the viral load dynamics do not qualitatively change within the time frame over which an infection typically occurs, and we continue to see stable viral loads.

### Many mutations with low fitness cost leads to between-host diversity in viral loads

To determine the expected distribution of spVLs among individuals predicted by our model, we first used numerical integration to determine the within-host dynamics of all possible $j$-type infections. For each infection-type, $j$, we could then calculate the mean viral load during chronic infection (our proxy for spVL), and in addition the infectivity of $i$-type virus at time $t$ since infection, $\beta_{ij}(t)$ for all $i$. Using this information we determined the distribution of $j$-type infections, and therefore spVLs, at equilibrium by using the next-generation matrix as defined in methods.

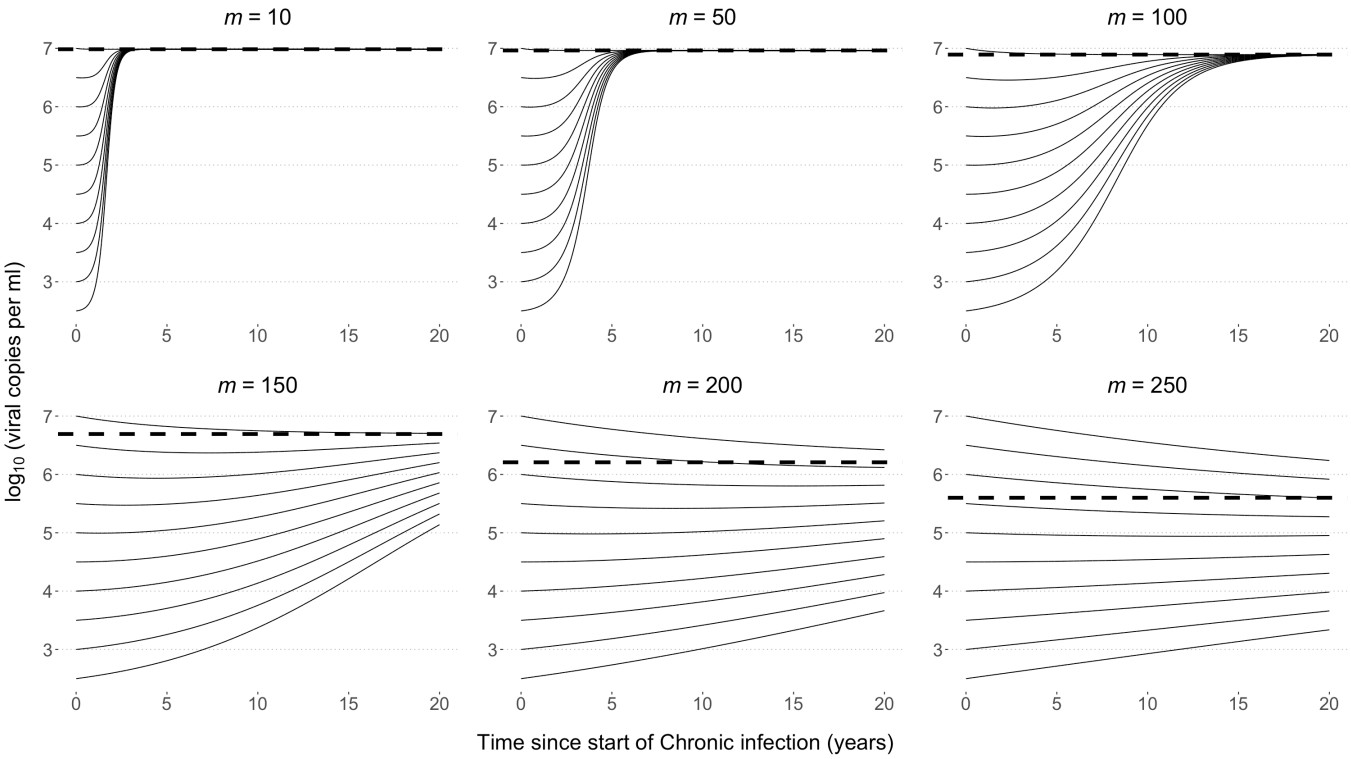

**Fig 2. Within host viral load trajectories.** The within-host viral load dynamics over time for 10, 50, 100, 150, 200 and 250 segregating sites, and varying initial numbers of mutations. The viral load of the equilibrium solution is shown by the black dashed horizontal line. The viral load changes as the virus population evolves over time, with new deleterious mutations appearing that are then lost due to purifying selection. When the cost of a mutation is high, viral loads climb to the maximum possible value. As we increase the number of segregating sites and reduce the fitness cost of a mutation, the selection against weaker virus types is balanced by the influx of mutations, and consequently the within-host dynamics are extremely slow relative to the typical duration of chronic infection.

The distribution of spVLs varies substantially as we increase the number of segregating sites (Fig 3A–F). As the number of segregating sites increases, the average viral load at endemic equilibrium across individuals falls to the range of previously reported average spVLs, with 5.14, 4.78, and 4.562 (viral copies per ml) for 150, 200 and 250 segregating

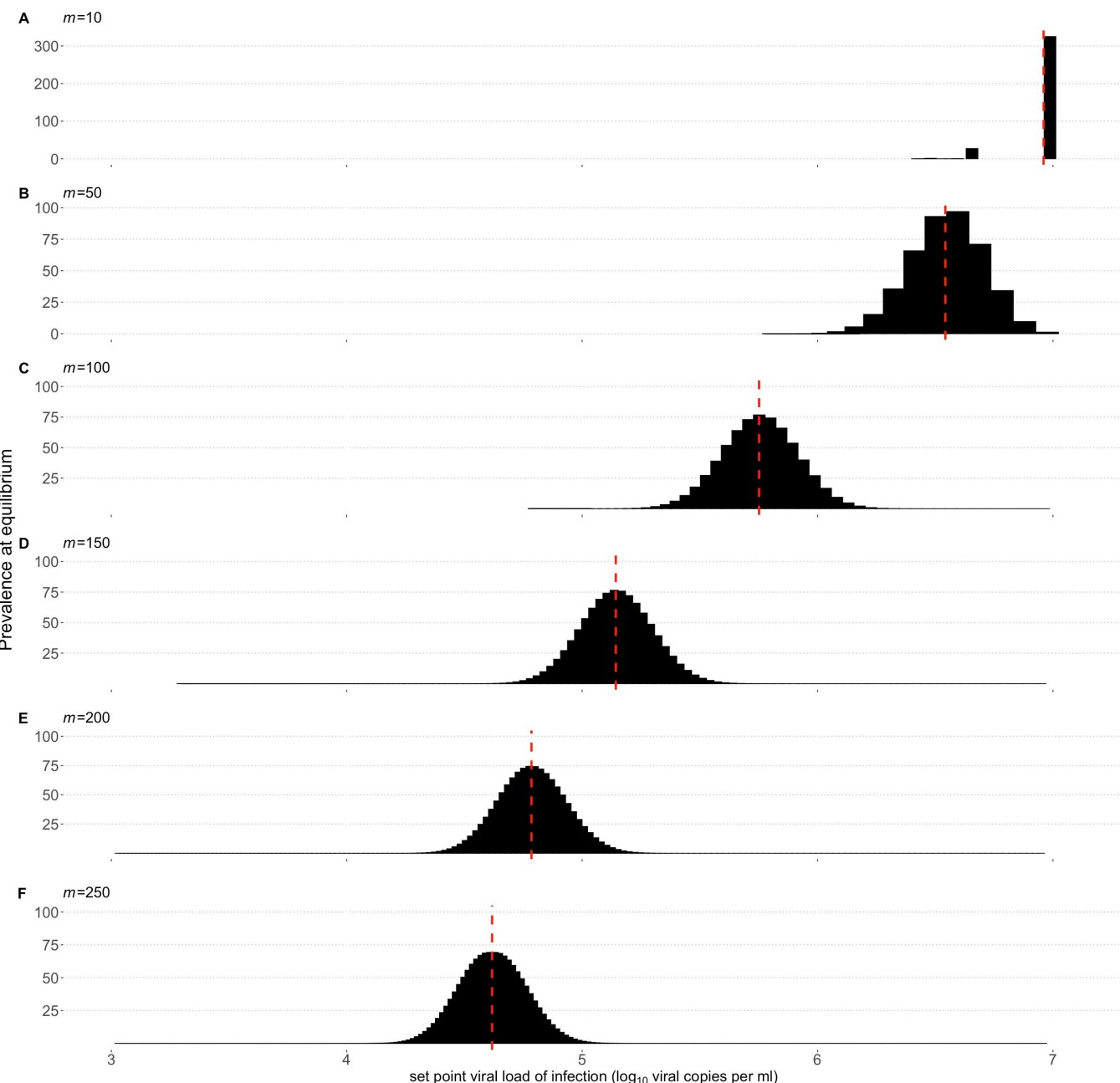

**Fig 3. Between-host outcomes at endemic equilibrium.** A) The distribution of set-point viral loads (spVL) at the between-host equilibrium. The fittest virus type rapidly dominates within-host and short-sighted evolution dominates. Red dashed lines denote the average within-host viral load. B-F) Greater within-host diversity provides a larger pool of variants that can be transmitted and on which between-host selection can act. As the number of segregating sites increases, the within-host dynamics slow and the virus is better able to evolve between-host towards an intermediate spVL that maximises transmission potential.

sites respectively (Fig 3D–F). The increased number of segregating sites, for which the associated fitness cost is lower, leads to slower within-host dynamics and more diverse populations, enabling the virus to evolve between hosts towards an intermediate spVL that maximises transmission potential. As a result, the epidemic size increases (S3 Fig).

Whilst having many weakly deleterious mutations resulted in little variation in viral load over the course of an individual infection, the spVL during chronic infection differed by an order of magnitude between individuals. However, the variation in spVLs was still less than observed in infected populations [18]. Host genetics are known to affect progression, and so we included host heterogeneity, such that the same infection type will induce different spVLs in individuals of different host type. For example, a host effect of 0.5 means that two individuals infected by the same virus type may differ in their spVL by a maximum of 1 $\log_{10}$ (viral copies per ml), 0.5 in either direction from the population mean. We find that in the case m = 250, spVLs at the endemic equilibrium are more realistically diverse when a host-effect is assumed, and the number of circulating virus types increases, and the between-host diversity in spVL increases with host effect as expected (Fig 4). We find the same effect on all choices of $m$ (supp. Fig 4).

### Many mutations with low fitness cost results in viral load evolving to intermediate levels between-hosts

To capture the epidemiological dynamics and how the mean spVL varies as the epidemic progresses, we numerically integrated the full nested model over the first 100 years of an epidemic. For each scenario (choice of $m$), three simulations were run where the average spVLs of infection types at the start of the epidemic differed and initially there is a single infected individual. Population average viral loads decrease or increase depending upon the initial viral load, with cumulative population changes occurring over approximately a century for m = 100 (Fig 5C), and over two centuries for larger m values (Fig 5D–E).

When we considered few mutations of larger effect, the fittest and most virulent variant rapidly outcompeted other virus types on the within-host scale. As a result, short-sighted evolution blocks between-host adaptation and the fittest variant dominates across the population at the expense of a reduction in transmission potential (Fig 5A–B). As the number of sites increase, the within-host dynamics slow and selection for transmission potential on the between-host scale drives the population-level dynamics, leading to gradual evolution towards an intermediate viral load. Importantly, we observe this dynamic for 100 segregating sites despite a high viral load at the point of mutation-selection balance. i.e., at within-host

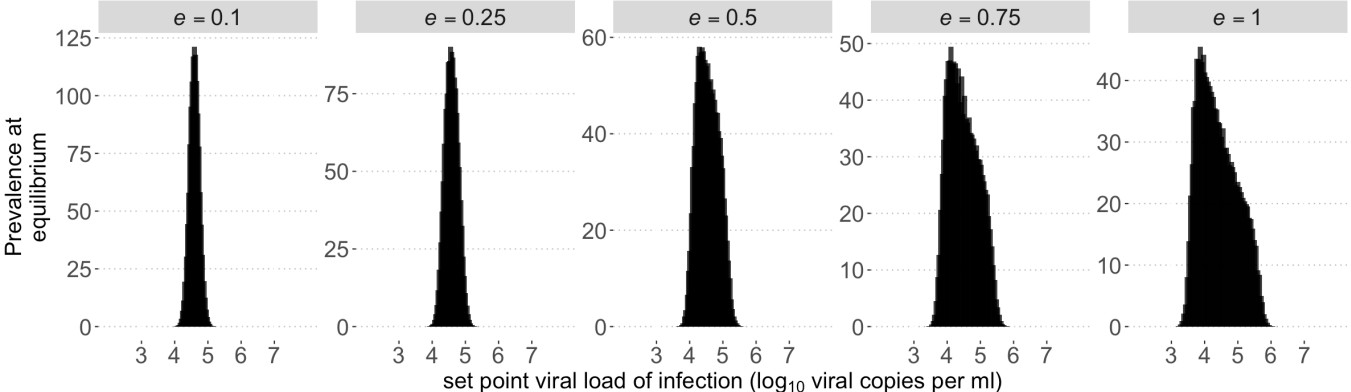

**Fig 4. Between-host outcomes at equilibrium for an increasingly heterogenous host population.** Histograms of spVLs in heterogenous infected population for different maximum host effect size, $e$, for 250 segregating sites. To account for the effect that host genetics has on viral load, we introduced a host specific additive effect to viral load. The size of the host effect is discretely uniformly distributed between $-e$ and $e$ and there are 50 host types. A maximum effect size of e = 0.1 (A) results in a small increase in the range of viral loads observed, and as we increase $e$ we observe a more realistic distribution of viral loads. Increasing the effect size towards e = 1 further flattens and skews the distribution. Corresponding results for other choices of m are present in supp. Fig 4.

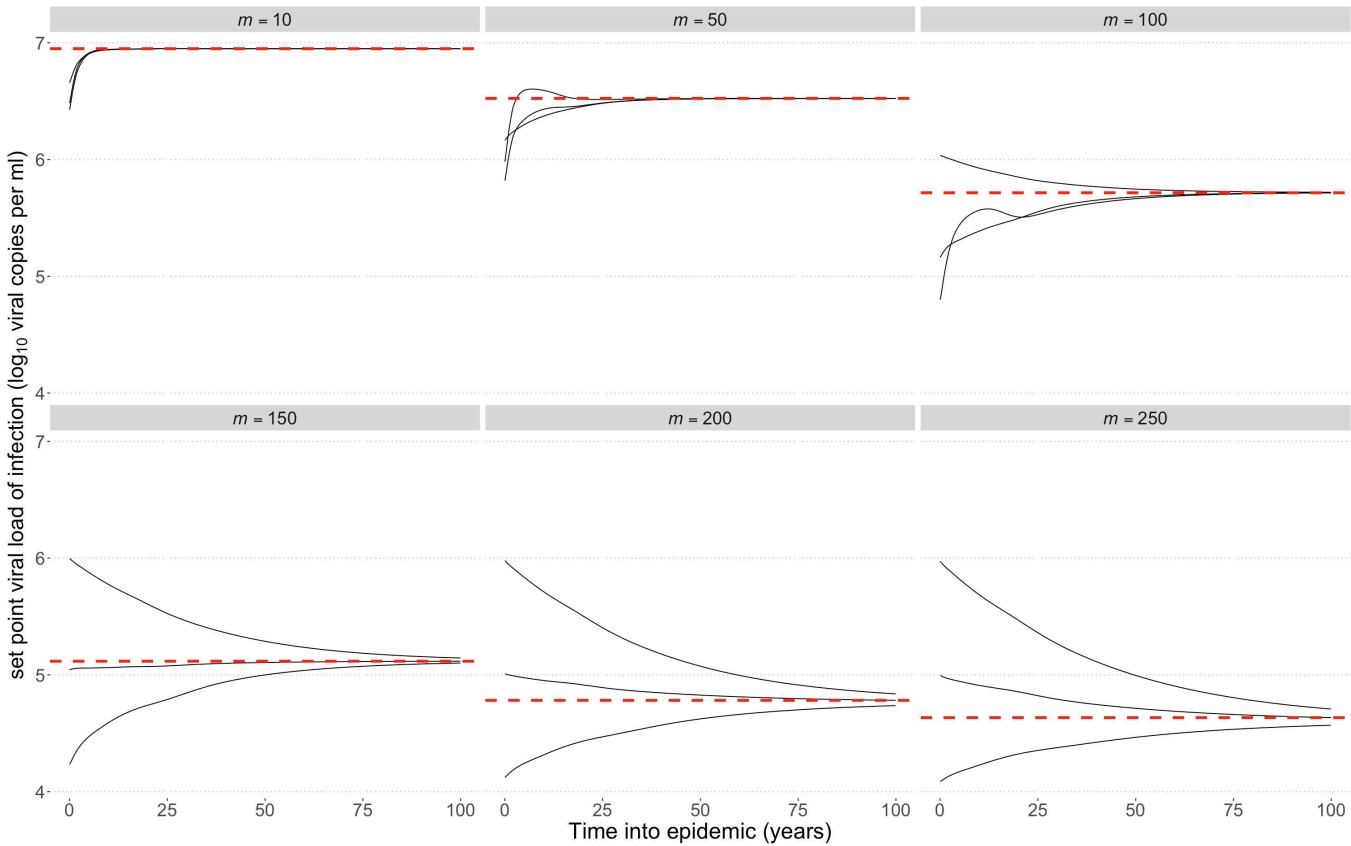

**Fig 5. Average spVL over time in simulated epidemics.** For $m = 10$, short-sighted evolution leads to the rapid dominance of the fittest virus type as the single circulating variant within a few years of the epidemic. For $m = 50$ and $m = 100$, the within-host dynamics create a distinctive pattern at the population level. When epidemics start with lower viral load viruses that have a relatively long infection duration, within-host evolution leads to the emergence and transmission of fitter variants with higher viral loads. This process accelerates the population-level increase in average spVL. When we assume a large number of weakly deleterious mutations ($m = 150, 200, 250$) within-host dynamics are sufficiently slow for selection for transmission potential to influence the epidemiological dynamics and lower the average spVLs, despite the comparatively higher within-host equilibrium viral load. We observe slow cumulative changes in the average spVL, with over 100 years taken for convergence at higher values of m.

equilibrium. In this case, the fitness cost of each mutation is sufficiently small to slow within-host adaptation, resulting in stable viral loads during the first years of transmission opportunity.

### Viral loads are similar within transmission pairs

Within-host evolution can cause significant genetic change in the quasispecies between early infection and the time of onward transmission. In the absence of host heterogeneity or other environmental effects, we expect estimates of heritability to reduce in response to increased within-host evolution [12]. We estimated heritability, $h^2$, by simulating transmission pairs in an infected population of homogeneous individuals at endemic equilibrium (Fig 6). Source infections were sampled based upon the infection-type population structure at the endemic steady state. Recipient infections were sampled based upon the transmission potential of each virus type during the infection of the source, and therefore the heritability estimates accounts for within-host evolution of viral factors.

We find heritability is lowest for few mutations of large effect ($m = 10$) due to rapid within-host evolution. In a homogenous population, all other scenarios have approximately equal heritability, likely due to the initial period of stability, high

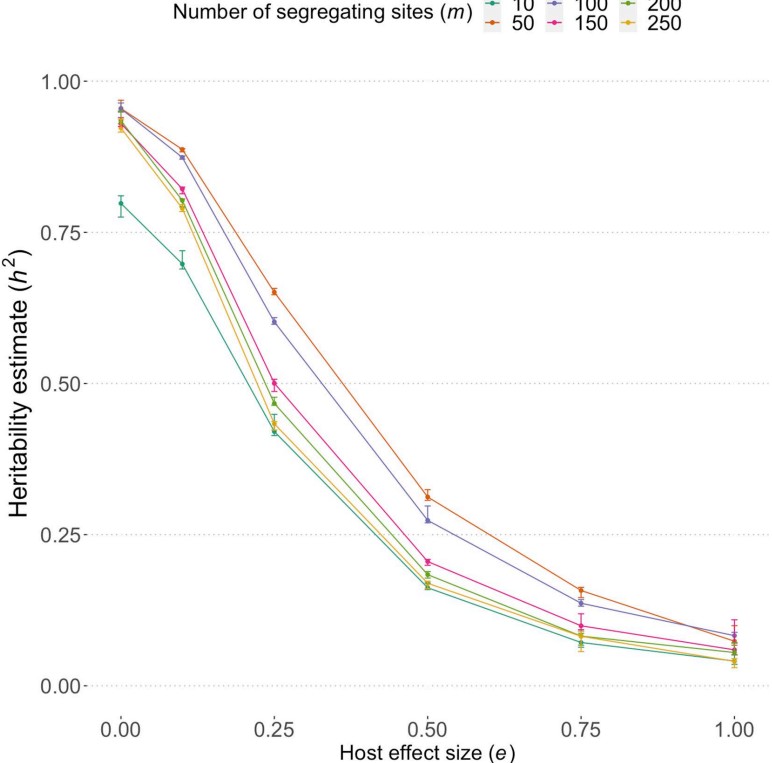

**Fig 6. Heritability estimates.** Heritability is estimated by a parent-offspring regression of spVL in simulated source and recipient pairs. The error bars indicate the standard deviation of the estimates taken from 1000 sampled sets of 500 transmission pairs. The infection type of the source (number of deleterious mutations of infecting virus type) is determined based upon the population-level prevalence of viral types at the endemic steady state. The virus type transmitted from source to recipient is determine based upon the transmission potential of each virus type over the course of the source infection. In a homogenous population, heritability is high, and naturally as a host-effect is introduced the amount of variability in spVL explained by the virus type, i.e., heritability, falls to within the range reported in multiple studies.

probability of transmission during acute infection, and the rapid changes occuring later into infection. As the host effect size increases, heritability decreases consistently across all scenarios, with the rate of decrease being similar regardless of the number of segregating sites, and heritability reduces to a range that has been reported in study estimates of broad-sense heritability. Heritability for $m=50$ and $m=100$ segregating sites is consistently greater than models with a larger number of segregating sites. This is because as the number of segregating sites increase, there is greater within-host variation in virus types, leading to more variation in transmitted viral loads between source and recipient.

## Discussion

There are several examples of mathematical models that predict pathogen dynamics by considering evolution across scales [40]. With a multi-strain model that nests within-host dynamics of HIV within a between-host model, Lythgoe *et al*. [32] found that within-host evolution is rapid enough that HIV should evolve to high levels of virulence at the expense of fewer onward transmissions, and so within-host evolution was a greater prevailing force than evolution between hosts. By incorporating a reservoir into the model, evolutionary processes are delayed, and short-sighted evolution is prevented; however, the impact of the reservoir was highly sensitive to its assumed size [41]. Van Dorp *et al*. [24] proposed an alternative model that considers immune escape and found that high host-heterogeneity in HLA types and consequently the escape and reversions of immune-escape mutations determine how spVL evolves. However, the model assumes that

mutations are time-separated and occur according to a Markov process, effectively limiting the tempo of within-host evolution. Here, we applied the nested model framework from Lythgoe, Pellis and Fraser [32] without a latent reservoir and consider whether many weakly deleterious mutations can provide an additional and more parsimonious mechanism for spVL evolution under two levels of selection. If the number of segregating sites is sufficiently large, we show this can reconcile multiple paradoxes in HIV biology: the stability of viral loads during chronic infection despite high mutation rates, the heritability of spVL despite considerable within-host evolution between transmission events, and the evolution of spVLs that maximise transmission. This mechanism could also explain why heritable viral factors determining viral load have been so difficult to identify, due to the high statistical power needed to detect them.

Alternative solutions to the problem of rapid within-host evolution have also been proposed, including the existence of rugged and complex fitness landscapes that are difficult for the within-host viral population to traverse [32], the cycling of virus through the (unreplicating) HIV reservoir which then slows the rate of within-host evolution [41] and viral factors that are heritable but are not under within-host selection, specifically polymorphisms that target the cell activation rate and are therefore beneficial to the entire virus population [42]. It is, however, difficult to reconcile the latter theory with the relationship between viral load and replicative capacity, and none of the described theories provide a fully satisfactory explanation of a heritable set-point viral load that varies by orders of magnitude between individuals, or why viral virulence factors have been so hard to identify [20].

A study of the fitness landscape of the HIV genome showed that for a substantial fraction of the genome mutations are weakly deleterious (cost <1%), particularly at synonymous sites but also at non-synonymous sites [38]. Synonymous mutations, while not directly affecting protein structure, can still impede viral fitness by influencing RNA stability, translation efficiency, or protein folding. The study also highlighted evidence that the deleterious component of the landscape is universal across infections and is fundamental to the high diversity within HIV group M diversity. In our modelling framework, it was not possible to explore beyond 250 segregating sites due to the significant computational demand of tracking within-host and between-host dynamics. While this represents only a subset of the possible mutational landscape, our qualitative findings are expected to hold if a larger number of sites were included. The scenarios explored here serve as a proof of principle, demonstrating how the accumulation of many weakly deleterious mutations can shape viral load distributions and evolutionary outcomes at both scales.

In this work, we show that the accumulation of deleterious mutations provides a parsimonious and biologically grounded explanation for several key features of HIV evolution: the high heritability of set-point viral load, the slow pace of change during chronic infection, and the apparent selection for intermediate transmission potential at the between-host scale. Our model demonstrates that purifying selection acting on many weakly deleterious mutations can constrain within-host adaptation, allowing population-level selection for transmission fitness to operate effectively.

An important additional mechanism shaping HIV evolution is the emergence of CTL escape mutations, which arise in response to host-specific HLA-mediated immune pressure [21,43–45]. These escape variants can confer a short-term replicative advantage within a host by evading immune detection, but often incur fitness costs that reduce viral replicative capacity, particularly when transmitted to hosts with different HLA backgrounds. This dynamic is reflected in the observation that viruses pre-adapted to a donor's HLA profile maintain higher viral loads when transmitted to recipients with similar HLA types—a phenomenon known as HLA footprints [31]. However, this mechanism cannot explain the dominant power of between-host evolution and the selection for maximal transmission fitness at a population level [24]. Additionally, HLA alleles are highly diverse at the population level, such that the effects of CTL escape mutations are diluted, limiting their long-term impact on spVL evolution.

By considering a host-effect on viral load we begin to explore the impact of host heterogeneity on heritability and between-host model outcomes, however the approach is heavily simplified compared to the complexities of HLA-specific CTL adaptions and their effect on viral fitness and viral load. Incorporating CTL escape into our framework would likely accelerate early within-host evolution and increase heterogeneity in SPVL across hosts, as the timing and nature of escape events are highly host-dependent. Over time, the accumulation of both escape and deleterious mutations could

further reduce viral fitness, potentially amplifying the effects predicted by our model. Conversely, compensatory mutations may partially offset these costs, complicating the relationship between genotype and phenotype. However, the impact of incorporating CTL escape into our modelling framework is difficult to predict, as it would depend on a range of interacting factors—such as the frequency and timing of escape mutations, the extent of HLA matching between transmission pairs, the magnitude of associated fitness costs, and the role of compensatory mutations.

While our model does not capture all sources of variation—such as rare large-effect mutations or the full complexity of virus-host interactions—it provides a tractable framework that reconciles the stability of viral load within individuals, the broad distribution of spVL among hosts, and the limited evolution of spVL at the population level. Future work that integrates both deleterious and beneficial mutations, including explicit modelling of CTL escape and HLA adaptation, may yield deeper insights into the evolutionary forces shaping HIV virulence and transmission.

As with all mathematical models, the model proposed here represents a significant simplification of complex biological processes for the sake of tractability and interpretation. Though this model reproduced known behaviour, in reality other processes will contribute to varying degrees, such as repeated immune escape and reversion across different host environments, and perhaps a small number of mutations of large effect. Developing an informed understanding of the virus factors that control virulence, and how they evolve in response to selection at the within- and between-host scales, will ultimately provide important insights into the severity of viral infections, including HIV, how this might change through time, and improve future treatments and public health policy.

## Supporting information

**S1 Fig. The frequency distribution of the viral population at equilibria.** A specific viral type is defined by its number of mutations. When we consider few mutations of large effect, the population is dominated by a single virus type of high relative fitness. As we increase the number of segregating sites and lower the associated fitness cost, the population becomes increasingly diverse, which has implications for the viral variants that are transmitted and between-host evolution. This diversity in the within-host viral population creates a broader pool of viral variants available for transmission, potentially influencing both transmission dynamics and the trajectory of between-host evolution.
(TIFF)

**S2 Fig. Within host viral load trajectories for varying $s$ and $m$ = 100.** The within-host viral load dynamics over time for 100 segregating sites, varying $s$ ($s = 0.5 \times 10^{-2.9}$, $s = 10^{-2.9}$ $s = 2 \times 10^{-2.9}$) and varying initial numbers of mutations. In the main results, $s$ is fixed at $m = 100$. The viral load of the equilibrium solution is shown by the black dashed horizontal line. As the size of the selection coefficient, $s$, increases, the tempo of within-host evolution is faster and the average viral load at the equilibrium (black dashed line) increases.
(TIFF)

**S3 Fig. The total prevalence at the endemic steady state.** As the number of segregating sites increases, the within-host dynamics slow down and the viral population is more diverse. As a result, between-host selection is able to select the virus types with greatest transmission potential, ultimately increasing the endemic prevalence.
(TIFF)

**S4 Fig. Between-host outcomes at equilibrium for an increasingly heterogenous host population for all models.** Histograms of spVLs in heterogenous infected population for different maximum host effect size, $e$, for A) 50 B) 100, C) 150 and D) 200 segregating sites. To account for the effect that host genetics has on viral load, we introduced a host specific additive effect to viral load. The size of the host effect is discretely uniformly distributed between $-e$ and $e$ and there are 50 host types. Increasing the host-effect broadens the distribution of spVLs.
(TIFF)

## Author contributions

**Conceptualization:** Christophe Fraser, Katrina Lythgoe.

**Formal analysis:** Harriet Longley.

**Investigation:** Harriet Longley.

**Methodology:** Harriet Longley, Chris Wymant, Katrina Lythgoe.

**Supervision:** Christophe Fraser, Katrina Lythgoe.

**Visualization:** Harriet Longley.

**Writing – original draft:** Harriet Longley.

**Writing – review & editing:** Christophe Fraser, Chris Wymant, Katrina Lythgoe.

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
