## [Decision Letter · Decision Letter 0]

7 Jul 2025

Attenuation of HIV severity by slightly deleterious mutations can explain the long-term trajectory of virulence evolution.

PLOS Computational Biology

Dear Dr. Longley,

Thank you for submitting your manuscript to PLOS Computational Biology. After careful consideration, we feel that it has merit but does not fully meet PLOS Computational Biology's publication criteria as it currently stands. Therefore, we invite you to submit a revised version of the manuscript that addresses the points raised during the review process.

Please submit your revised manuscript within 60 days Sep 06 2025 11:59PM. If you will need more time than this to complete your revisions, please reply to this message or contact the journal office at ploscompbiol@plos.org. Please include the following items when submitting your revised manuscript:

We look forward to receiving your revised manuscript.

Kind regards,

Alexandre V. Morozov, Ph.D.

Academic Editor

PLOS Computational Biology

Natalia Komarova

Section Editor

PLOS Computational Biology

**Journal Requirements:**

4) Please amend your detailed Financial Disclosure statement. This is published with the article. It must therefore be completed in full sentences and contain the exact wording you wish to be published.

2) If any authors received a salary from any of your funders, please state which authors and which funders..

**Reviewers' comments:**

Reviewer's Responses to Questions

**Comments to the Authors:**

Reviewer #1: In their paper "Attenuation of HIV severity by slightly deleterious mutations can explain the long-term

trajectory of virulence evolution", Longley et al propose a new explanation for the lack of specific mutations that are associated ith set-point viral loads. According to their hypothesis many slightly deleterious mutations could be responsible for that. This could also explain the wide variation is set-point viral loads across infected individuals.

This paper presents an original and appealing hypothesis for some of the remaining discrepancies in our understanding of the evolution of the set-point virus load of HIV. To assess its validity, the authors developed a computational model that takes into account the within-host dynamics, diversification and selection of HIV, and its epidemiological transmission. While the resulting model is fairly complex, it is built on previous modeling work (by Lythgoe, Pellis and Fraser) and its parameters are empirically well supported. I have been reviewing this work a few weeks ago in the context of the viva of the first author. I already liked it the first time I read it: an inspiring conceptual idea assessed by well-executed modeling and analysis. I therefore have only minor points to raise.

My first point relates to the motivation of the study: in short the authors say that the heritability of the set-point viral load is between 20-30% (l.63), but the genetic basis for this remains unclear because:

"Specific viral mutations with large effects on fitness have been identified (44), but these are relatively low in number and the total narrow-sense heritability from all genetic hits is significantly lower than our estimates of broad-sense heritability." (l.520)

Ref 44 - Gabrielaite et al - estimates that four viral mutations account for 8.2% of the variation in the set-point viral load. While this does not add up to 20%, it is consistent with broad-sense heritability estimates by Hodcroft et al (5.7% - not cited by the authors) and Bertels et al (8% - cited) that assumed neutral (Brownian) trait evolution. (In my view, these estimates were lower than previous estimates because they were adjusted for covariates such as age and sex.) Assuming stabilizing ("Ornstein-Uhlenbeck") selection, the heritability estimates are higher (e.g. 29% in Bertels et al), and this type of selection has more statistical support than neutral evolution of the set-point virus load.

It would be great if the authors could discuss briefly if the narrow-sense heritabilities estimated by GWAS should be compared to the estimates based on neutral or stabilizing trait evolution. GWAS usually adjust for demographic covariates, so their heritability estimates should be compared to adjusted broad-sense estimates. But I am unclear about the assumptions about trait evolution underlying GWAS. This is important as it would tell us exact how far off the different heritability estimates are.

It would be great if the authors could specify the mutation matrix qij. They state on l.133 that "each newly infected cell can acquire or lose at most one mutation". Does that mean that going from the reference with no mutation to a single mutation happens with the genomic mutation rate 3x10^-5*10,000 = 0.3 while loosing that specific mutation happens with 3x10^-5*m? (What confuses me is that there is m single-point mutants in class x_1, choose(m,2) in x_2, etc, so forward and backward mutation rates between virus type i and j are different, vary by i, and depend on the number of segregating sites.)

The authors simulate with varying numbers of segregating sites from 10 to 250 and use the landscape of mutation costs estimated by Zanini et al (ref 36). Could the authors state how many segregating site Zanini et al observed in their data and the average cost they found for comparison with the authors' modeling assumptions? (It could even be plotted into Fig. 1A.)

Lastly, it would be great if the authors could comment on how CTL escape mutations, that often occur during HIV infection and have large beneficial fitness effects, would affect their results. The authors discuss how the HLA alleles of hosts might affect the set-point virus load evolution. Would this render deleterious mutations less effective in slowing within-host evolution, or the reverse? (I am aware that the authors discuss van Dorp et al's work provides an alternative hypothesis for the evolution of the set-point viral load of HIV. I am basically asking what happens if you combine the two hypotheses.)

Very minor points and typos:

- Fig. 1,2,3,4,5 do not require color

- there are still instances of "I" instead of "we", e.g. "I applied" in l. 495.

Reviewer #2: Plos Comp Bio June 2025

I love the main idea of this paper, to try to determine if many mutations of small deleterious effect could explain the variation in viral loads and the heritability of viral loads in HIV.

I enjoyed thinking about the model you proposed. I imagine that a large number of sites where mutations are slightly deleterious creates genetic variation within the host (quasi species or mutation selection balance). The resulting viral load influences the length of infection and also infectivity. I guess you are suggesting that some deleterious mutations and therefore lower VL would be good for the virus in the long run because it lengthens the live of the host and therefore increases the chance of infecting a new host. Then, from the within-host variation, a random viral particle infects the new host. If within host variation is very large, then heritability will be low, but if within host variation is moderate, we could get some heritability and also between host variation. In addition, to keep some between host variation, you need the dynamics of reaching the mutation-selection equilibrium to be slow (if fast, every host would have the same VL within a year). To get slow dynamics, you need small selection coefficients. To get within and between host variation you need a high number of sites that can mutate.

I think that your model would also work if you added a range of selection coefficients for the sites. There is no need to have only one s value.

While I think the paper is interesting and could be important for the field, I have some serious concerns.

1. I don’t like that you vary the number of sites where mutations can occur (m) with the selection coefficient of the mutations (s), as shown in figure 1A. [By the way, fig 1A should maybe be in the methods? It is an assumption, not a result).

The reason I don’t like it is because it doesn’t allow you to see the effect of m and s separately. The selection coefficient s should determine the time it takes until an equilibrium is reached. The number of mutations m should determine the average fitness of the population at equilibrium.

I would love to see a series of heatmaps (or similar) that show values of m on the x-axis, values of s on the y-axis and then on the z-axis the average VL, the between host variation in VL, and the heritability of VL.

2. Your result in figure 1B, that more mutations lead to lower fitness, independent of the selection coefficient is a well known result in population genetics. See for example equations 1.18 - 1.20 in Joachim Permissions pop gen notes:

https://www.mabs.at/fileadmin/user_upload/p_mabs/2024-lecturenotes.pdf

[You can find this in many books too, but these notes are free to access].

It would be good if you mention the term “mutational load” which is what population geneticists call this effect. It’d also be good to check if your model still has this behavior if you change s but not m.

3. Somewhere in the results, you decide that m = 250 is the best fitting parameter choice. I am not entirely convinced by that, because it seems to me that this depends a lot on your choice of s and your choice of the max VL.

4. I am not sure I am convinced of adding a host effect to explain the variation in VL among hosts. I guess you need to find parameter values that lead to enough variation between hosts and also sufficient heritability. I guess your paper is meant to be a proof of concept, and not a formal fitting of the model to available data, but maybe you can explain your choices a little more here. Currently it feels a little ad hoc.

5. Would it be possible to show some zoomed in results from your simulations? Like for a given host, what is the behavior of individual mutations? How does the VL behave in that host? This is just out of curiosity & to get a better sense of what is happening at the host level and how that influences the epidemic level?

Small things:

Line 30-32: This statement seems a little strong in my opinion because only one (or two) paper have shown this.

Line 34-36: I don’t understand this sentence.

Line 41: trade-offs. I think there is a new review on trade offs out of Brandon Ogbunu’s lab. May be worth citing here. Also: here you argue that it is hard to proof a trade-off (I agree) which contradicts with the strong statement in the first sentence of the Introduction).

Line 64: “Most part unexplained” — this makes me curious, which part *is* explained?

Line 69-71: I don’t understand this sentence.

Line 82-94: I found this paragraph somewhat hard to follow.

Line 111: “segregating sites” This was confusing to me because they are potentially segregating, but not necessarily, right? Maybe just say: sites.

Line 121: the quasi species equation is the same as mutation-selection balance, right?

Line 136: something odd with a footnote at the end of the line.

Line 153: I don’t understand why you need to track all possible combinations of viral variants. Not a big deal, just curious.

Line 181: lambda = 5 / m . Why 5? I am confused!

Line 194: hill should be Hill

Line 283: “I” should be “we”?

Line 303: “effectively neutral” I don’t understand that statement.

Line 360: “equilibrium is reached” I don’t agree with that. The figures don’t look like an equilibrium is reached.

Line 374: “next generation framework” I am not sure what that means here.

Line 380: here you say that increased m leads to slower dynamics, but it is actually decreased s that does this.

Figure 4: I suggest using yellows for this figure to show that it is linked to the m = 250 case of the previous figure.

Line 437: “Short sighted” I am not sure why this is considered short sighted. Is it because it doesn’t maximize infectivity? Now that I am typing this up, I guess I am not sure how infectivity is affected in your model by m and s.

**Have the authors made all data and (if applicable) computational code underlying the findings in their manuscript fully available?**

Reviewer #1: None

Reviewer #2: Yes

PLOS authors have the option to publish the peer review history of their article (what does this mean? ). If published, this will include your full peer review and any attached files.

**Do you want your identity to be public for this peer review?** For information about this choice, including consent withdrawal, please see our Privacy Policy .

Reviewer #1: No

Reviewer #2: **Yes: ** Pleuni Pennings

**Figure resubmission:**
---

## [Decision Letter · Decision Letter 1]

14 Nov 2025

Dear Miss Longley,

We are pleased to inform you that your manuscript 'Attenuation of HIV severity by slightly deleterious mutations can explain the long-term trajectory of virulence evolution.' has been provisionally accepted for publication in PLOS Computational Biology.

Also, please address the few minor suggestions raised by the reviewers in response to the revised manuscript.

Best regards,

Alexandre V. Morozov, Ph.D.

Academic Editor

PLOS Computational Biology

Natalia Komarova

Section Editor

PLOS Computational Biology

Reviewer's Responses to Questions

**Comments to the Authors:**

Reviewer #1: The authors addressed all the points I raised very well in the response and their revised manuscript.

There seems to be typos in the definition of qij in lines 181-183: the index should not be "j+1,j" in 2. and 3., but "j-1,j" and "j,j" if I understand correctly.

Reviewer #2: I’ve read the response of the authors and teh new manuscript. I like it, and I hope it will lead to more research on the topic.

I wonder if it would be useful to think about the concept of mutational load which is the average loss of fitness due to mutation in a mutation-selection balance situation. Your results in figure 3 fit very well with this concept.

https://www.blackwellpublishing.com/ridley/a-z/Mutational_load.asp#:~:text=Mutational%20load%20is%20the%20total,and%20its%20production%20by%20mutation.

I wonder also if it is worth mentioning that M184V typically reverts back to WT within a few months (example of highly deleterious mutation that would not contribute much to heritability) whereas other mutations such as the NNRTI resistance mutations typically take longer to revert, and they are therefore more common in the population and will contribute more to heritability.

https://pubmed.ncbi.nlm.nih.gov/21451005/

**Have the authors made all data and (if applicable) computational code underlying the findings in their manuscript fully available?**

Reviewer #1: None

Reviewer #2: None

PLOS authors have the option to publish the peer review history of their article (what does this mean? ). If published, this will include your full peer review and any attached files.

**Do you want your identity to be public for this peer review?** For information about this choice, including consent withdrawal, please see our Privacy Policy .

Reviewer #1: **Yes: ** Roland R Regoes

Reviewer #2: No

---

## [Editor Report · Acceptance letter]

PCOMPBIOL-D-25-00838R1

Attenuation of HIV severity by slightly deleterious mutations can explain the long-term trajectory of virulence evolution.

Dear Dr Longley,

I am pleased to inform you that your manuscript has been formally accepted for publication in PLOS Computational Biology. Your manuscript is now with our production department and you will be notified of the publication date in due course.

With kind regards,

Judit Kozma
